# Development and evaluation of a stepwise clinical competency development program for new nurses: A single-group repeated-measures quasi-experimental study

Shinhye Ahn[1], Hye Won Jeong[2]*

**1** Department of Nursing, Sunchon National University, Suncheon-si, Jeollanam-do, Republic of Korea, **2** Department of Nursing, Korea National University of Transportation, Jeungpyeong-gun, Chungcheongbuk-do, Republic of Korea

* hyewon1129@ut.ac.kr

## Abstract

### Background

New nurses often encounter adaptation challenges in hospital settings owing to gaps in clinical knowledge and skills, leading to high turnover rates and patient safety concerns. Although various orientation and preceptorship programs exist, they remain inconsistent and rarely evaluated longitudinally. Effective training is essential to support clinical adaptation. This study aimed to develop a structured, stepwise clinical competency development program (SCCDP) for new nurses and evaluate its effectiveness.

### Methods

This study employed a single-group repeated-measures quasi-experimental design. From September 2023 to July 2024, 49 new nurses from C University Hospital in South Korea participated in the SCCDP, which consisted of lectures, practice, and simulation validated by experts. The outcomes measured included basic and advanced clinical knowledge, clinical performance ability, and self-efficacy at four intervals: pre-intervention, immediately post-intervention, and at three and six months post intervention. Data were analyzed using repeated-measures and Friedman's ANOVA, and effect sizes were calculated.

### Results

The SCCDP significantly improved new nurses' basic clinical knowledge immediately post-intervention and three and six months post-completion ($F = 40.01$, $p < .001$), with the largest effect size observed in medical device operation (ES = 1.56–1.65). Advanced clinical knowledge also demonstrated significant enhancement across

**Data availability statement:** All relevant data are within the paper and its Supporting information files.

**Funding:** This study was funded by the National Research Foundation of Korea (NRF-2022R1F1A1067574).

**Competing interests:** The authors have declared that no competing interests exist.

all time points (F = 26.06, $p < .001$), with the greatest increase occurring immediately after the SCCDP (ES = 1.84), particularly in emergency nursing (ES = 1.22). Clinical performance ability showed notable gains at three and six months post-program ($\chi^2 = 55.92$, $p < .001$), with the most substantial improvement in interpersonal and communication skills (ES = 2.05–2.09). However, self-efficacy did not change significantly over time (F = 2.80, $p = .066$).

## Conclusions

The SCCDP enhanced new nurses' knowledge and clinical performance and demonstrated sustained effects over time. These findings support the implementation of structured, competency-based education to facilitate new nurses' adaptation and retention in clinical practice.

---

## Introduction

The turnover rate among new nurses with less than one year of employment following their 2022 appointments was 52.8%, and this trend continues to rise [1]. New nurses experience challenges in adapting to the hospital's organizational environment, often due to insufficient knowledge and professional skills, excessive workload, and difficulties in communication and collaboration with other medical staff and colleagues [2]. Nurse turnover leads to significant financial losses for hospitals [3], with new nurse turnover comprising the largest portion of the overall nursing attrition [1], which further increases the burden on patients and remaining staff. Although previous research highlights the need to support new nurses' clinical adaptation through education and structured training [4,5], existing training programs are often inconsistent, insufficiently standardized, and lack long-term evaluations of effectiveness [6,7]. Thus, developing a structured, evidence-based educational intervention is crucial to systematically address the competency gaps and adaptation challenges faced by new nurses, ultimately aiming to reduce turnover and improve patient care outcomes [8,9].

To address the limitations of preceptorship and the previous education system for new nurses and to support new nurses' adaption to clinical practice, there has been a call to implement an effective educational system and deploy dedicated training personnel [10]. The role of clinical nurse educators is gaining prevalence worldwide, as they provide quality education and support new nurses in their transition into confident, competent practitioners [11]. To establish a systematic, practice-oriented curriculum and training staff that reflect the unique characteristics of the working environment and healthcare institutions [12], the Ministry of Health and Welfare and the Korean Nurses Association launched a support initiative in 2019 for clinical nurse educators and training programs for new nurses, accompanied by the release of management system guidelines [13]. Clinical nurse educators, primarily deployed at national and public hospitals, provide essential job training, identify the clinical difficulties experienced by new nurses, develop practical educational programs,

and deliver standardized training for complex nursing skills [13]. However, the current circumstances of Korean medical institutions, which include a widely varied training period and curriculum for new nurses across institutions and insufficient deployment of clinical nurse educators, limited their ability to fully support new nurses in adapting to the clinical environment [12,14].

Among the various factors influencing turnover rates perceived by new nurses, work intensity was identified as the most significant, followed by practical competency, which accounted for approximately 20.5% [15]. Insufficient preparation for practice can lead to transition shock, underscoring the need for clinical nursing education that supports the transition process in clinical settings [16]. Education and effective training that foster critical thinking skills are essential for enhancing the clinical practice competency of new nurses [17]. Providing recurring experiences closely aligns with clinical practice, along with education and constructive feedback tailored to the competency levels of new nurses, which has been shown to improve their practical skills [16]. To strengthen new nurses' practical skills, support their adaptation to the clinical setting, and reduce turnover associated with insufficient competency, it is necessary to develop and implement a practical competency enhancement program for them led by dedicated clinical nurse educators.

As new nurses encounter increasing challenges during their transition from beginners to fully qualified practitioners, a step-by-step education and development strategy must be established that considers their tenure and professional growth processes [9]. In the context of this study, inconsistent training approaches across different hospitals, insufficient practical competency preparation, and variability in preceptorship quality have been identified as critical issues hindering new nurses' clinical adaptation [18]. Furthermore, inadequate competency among new nurses and high turnover rates have been linked to increased patient safety risks, lower quality of patient care, and greater dissatisfaction among both patients and healthcare staff [19]. Accordingly, this study aimed to develop and implement a stepwise clinical competency development program (SCCDP) tailored to the tenure of new nurses, assess its effectiveness, and provide foundational data for a standardized integrated adaptation support program that can be applied to new nurses in the future. This study differs from previous research by employing a longitudinal repeated-measures design and a competency-based framework that enables systematic evaluation of both basic and advanced clinical competencies over six months. Ultimately, this approach is expected to enhance the quality of patient care and improve job satisfaction by supporting new nurses to strengthen their competencies and adapt to clinical environments.

## Conceptual framework

The SCCDP for new nurses was structured based on the theoretical content of competency-based education (CBE) (Fig 1). CBE is an educational approach focused on ensuring that learners acquire specific competencies and effectively apply them to real-life situations [20]. This conceptual framework is designed to enhance the clinical adaptation and competency of new nurses and incorporates several key elements. At the core of the SCCDP is a step-by-step practical competency-strengthening process that integrates the CBE principles. The SCCDP is divided into basic and advanced stages, each structured around clearly defined competency goals. This approach facilitates the gradual development of competencies among new nurses, enabling flexible education tailored to their learning levels [21].

Competency-based assessment, the core principle of CBE, is another key element of this framework. This evaluation method continuously monitors the learning progress of new nurses and measures their performance in real clinical situations, extending beyond mere knowledge acquisition [22]. Evaluations were conducted before the SCCDP, immediately after, and three and six months after to track changes in competency over time. This approach enabled a comprehensive assessment of both the short- and long-term effects of the SCCDP. To evaluate the SCCDP's effectiveness, various outcome variables, including basic and advanced clinical knowledge, clinical performance ability, and self-efficacy, were measured. Basic and advanced clinical knowledge evaluated the theoretical foundation of nursing and the understanding of complex clinical situations, whereas clinical performance measured the ability to apply nursing skills in real-world settings. Self-efficacy was used to assess new nurses' confidence in performing nursing tasks. This multifaceted assessment

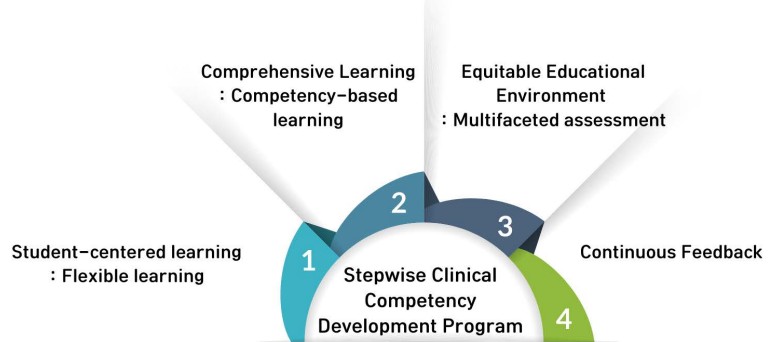

Legend. The conceptual framework illustrates how the stepwise clinical competency development program integrates the principles of competency–based education. It emphasizes student–centered and flexible learning, comprehensive and equitable competency development, and a multifaceted environment designed to enhance new nurses' clinical adaptation and professional competency.

**Fig 1. Conceptual framework of the SCCDP grounded in competency-based education.** This figure presents the conceptual framework of the SCCDP grounded in competency-based education. The framework illustrates the stepwise progression of clinical competency development from the basic stage to the advanced stage for new nurses. Core components include structured educational methods (lectures, practice, and simulation), competency-based assessment, and continuous feedback. Outcomes of the program are evaluated longitudinally at four time points (pre-intervention, immediately post-intervention, and at three and six months post-intervention) to assess changes in basic and advanced clinical knowledge, clinical performance ability, and self-efficacy.

allowed us to comprehensively measure the overall competency development in new nurses [23]. Continuous feedback, a key component of this framework and an important feature of CBE, helps individuals recognize their strengths and areas for improvement, thus enabling goal-oriented learning [21].

Finally, the framework was designed to be flexible and adaptable. The ability to adjust programs to meet new competency requirements or respond to changes in the healthcare environment is an important feature of the rapidly evolving field of healthcare [21]. This flexibility allows the SCCDP to continuously incorporate the latest medical practices and educational methodologies. This conceptual framework aimed to bridge the gap between theory and practice and promote effective clinical adaptation by applying CBE principles to the nursing education of new nurses. The SCCDP's long-term effectiveness can be assessed by tracking changes in competency over time, contributing to its continuous improvement and optimization.

## Materials and methods

### Research design

This study employed a single-group, repeated-measures quasi-experimental design to assess the impact of the SCCDP on new nurses' basic and advanced practical knowledge, clinical performance ability, and self-efficacy. This research was conducted at a single hospital, aiming to provide structured, systematic education for newly recruited nurses joining the institution. Due to ethical considerations and the necessity of providing standardized training to all new nurses at designated intervals, it was not feasible to include a control group in this study.

### Participants

In this study, new nurses were operationally defined as individuals with less than one year of clinical experience, including any prior clinical experience at other hospitals, who were newly employed at a university hospital in Korea between August and December 2023. Individuals with total clinical experience exceeding one year (including prior experience at

other institutions) participated in the SCCDP but were excluded from the data analysis. The inclusion criteria included those who provided consent for this study and completed the entire training. Those who partially completed the SCCDP or withdrew from the study were excluded from the analyses. The sample size was calculated using the G*Power 3.1.9.7 software. A medium effect size ($f = 0.25$), based on the effect size reported by Kim et al. [24], was applied with a significance level of .05 set for a single-group repeated measures analysis of variance (ANOVA). With a power of .95, the minimum sample size required was 43. To account for the anticipated dropout rate of approximately 20%, 50 participants were initially recruited. Since one participant dropped out due to resignation, the final sample size was 49.

## Program development

The SCCDP was divided into two stages, basic and advanced, with each stage structured around clearly defined competencies required for new nurses to progress from beginner to practitioner levels (Table 1). Each SCCDP session was developed using theoretical content, practice, and simulation-based education methods to accommodate diverse learning styles and facilitate the evaluation of practical competencies (Fig 2). Expert validation of the SCCDP's overall content and structure was conducted by three nursing school professors, two clinical nurse educators, and two nursing managers with master's degrees or higher, resulting in a content validity index (CVI) of .95. Additionally, the specific educational materials used during the training sessions (e.g., manuals, teaching materials, simulation scenarios) were separately reviewed and validated by the same expert panel to ensure content accuracy, clinical relevance, and appropriateness for the educational objectives.

The lectures and practical sessions were delivered by two clinical nurse educators with more than seven years of experience, supported by five assistant educators. Feedback on individual learning needs was provided by the same educators. During their first eight weeks of employment, new nurses participated in the SCCDP's basic phase alongside a preceptorship course. The basic stage were held on designated education days, during which new nurses were not assigned direct patient care responsibilities, although their preceptors continued to manage patient care while simultaneously educating and supervising the new nurses. This phase included four hours of formal training sessions held every two weeks. The basic phase covered foundational clinical skills and knowledge, such as medication administration, proper use of medical devices, and infection control. Additionally, it comprehensively addressed essential competencies for the hospital environment, including Situation, Background, Assessment, Recommendation (SBAR)-based handover training, communication, and teamwork.

Four months after joining the hospital, new nurses transitioned to the advanced stage of SCCDP. This stage consisted of four sessions, each lasting 8 hours and held at 2-month intervals, covering emergency nursing, respiratory system nursing, renal system nursing, and circulatory system nursing. These advanced sessions were conducted on the hospital's official education off-days, with each session lasting eight hours. The advanced stage emphasized enhancing decision-making and critical thinking skills in complex clinical situations by integrating theoretical education and clinical scenario-based simulations. A key feature of this stage was the provision of immediate and specific feedback following each session to support learners' continuous development.

Practical components in this stage utilized action learning strategies, including team-based problem-solving and reflective practices, to support leadership and critical thinking competencies. Throughout the SCCDP period, an online interactive bulletin board platform called Padlet (https://padlet.com) facilitated asynchronous communication, allowing new nurses to ask questions anytime and receive prompt feedback from educators.

Additionally, to address the individual needs of learners, regular interviews were conducted among new nurses, preceptors, and the clinical nurse educator, with additional support provided as necessary. For new nurses who required extra assistance, a dedicated training nurse provided one-on-one shadowing sessions at clinical sites. Clear learning objectives and competencies were outlined at each stage, and self-assessment tools were provided to help learners track their progress. Regular feedback from new nurses and preceptors was used to continually revise and enhance the SCCDP's content and structure. Table 1 provides detailed information about the content covered in each session.

**Table 1. Overview of curriculum content and teaching methods for SCCDP.**

| Phase | Session | Content | Time (min) | Methods |
|---|---|---|---|---|
| Basic | 1 | Course introduction | 30 | Lecture |
| | | Use of electronic medical record system | 120 | Practice |
| | | Intradermal injection | 30 | Practice |
| | | Subcutaneous injection | 30 | Practice |
| | | Insulin preparation | 30 | Practice |
| | 2 | Intravenous injection | 60 | Practice |
| | | Safe medication administration | 40 | Lecture |
| | | Medication dosage calculation (Basic) | 40 | Lecture & Practical |
| | | Medication administration simulation | 100 | Simulation |
| | 3 | Use of patient monitor and medical devices | 120 | Practice |
| | | Use of infusion pump | 120 | Practice |
| | 4 | Nursing care for maintaining and managing skin integrity | 100 | Lecture |
| | | Nursing care for patients with impaired skin integrity | 140 | Simulation |
| | 5 | Infection control | 50 | Lecture |
| | | Multidrug-resistant organisms | 50 | Lecture |
| | | Infection control | 140 | Simulation |
| | 6 | Blood transfusion nursing care | 40 | Lecture |
| | | Blood transfusion nursing care | 80 | Simulation |
| | | SBAR and handover | 120 | Lecture & Practical |
| Advanced | 1 | Medication dosage calculation (Advanced) | 40 | Lecture & Practical |
| | | Emergency nursing | 50 | Lecture |
| | | Use of a defibrillator | 80 | Practice |
| | | Endotracheal intubation | 100 | Practice |
| | | Team-based CPR | 120 | Simulation |
| | 2 | Oxygen therapy | 80 | Lecture |
| | | Non-invasive positive pressure ventilation | 80 | Lecture |
| | | Arterial blood gas analysis | 80 | Lecture |
| | | Respiratory nursing care | 160 | Simulation |
| | 3 | Fluid therapy | 80 | Lecture |
| | | Acute kidney injury | 100 | Lecture |
| | | Nursing care for dialysis patients | 80 | Lecture |
| | | Renal nursing care | 140 | Simulation |
| | 4 | Normal electrocardiogram interpretation | 60 | Lecture |
| | | Arrhythmia | 80 | Lecture |
| | | Pacemaker principles and management | 50 | Lecture |
| | | Hemodynamic monitoring | 80 | Lecture & Practical |
| | | Cardiovascular nursing care | 120 | Simulation |

*Note.* SCCDP = Stepwise Clinical Competency Development Program; SBAR = Situation, Background, Assessment, Recommendation; CPR = Cardiopulmonary resuscitation.

## Data collection

The data collection period for this study spanned from 4 September 2023–31 July 2024, with data collected at four points: before the SCCDP began, immediately after program completion, and at three and six months following program completion. Prior to the start of the SCCDP, permission was obtained from the nursing department of the study hospital to post

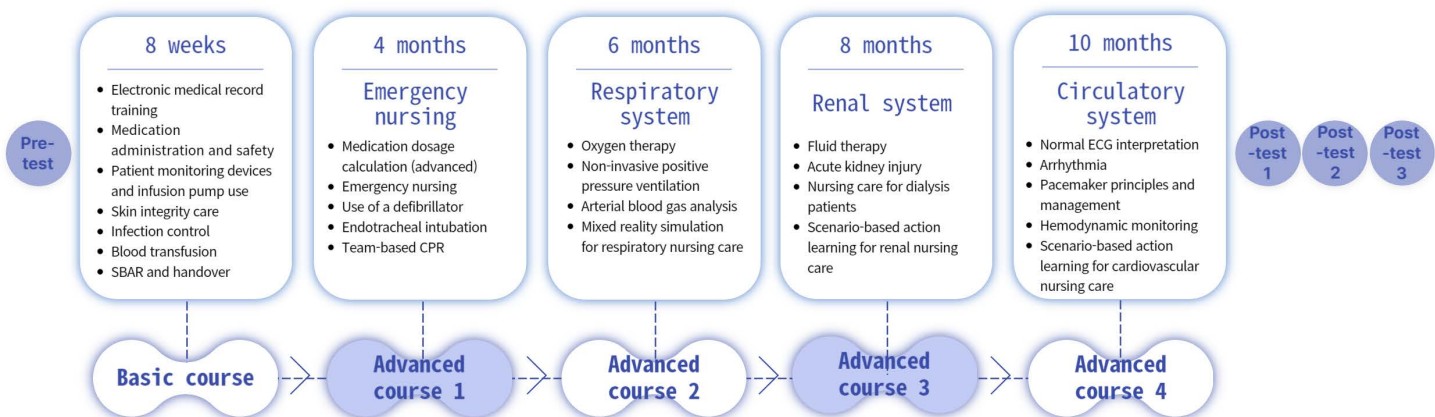

Legend. The figure presents the timeline and structure of the SCCDP, consisting of an 8-week basic course and four advanced courses over six months. Each phase integrates lectures, practice, and simulations addressing essential competencies such as medication administration, emergency nursing, respiratory and renal care, and team-based CPR. Post-tests were conducted immediately after, three months, and six months post-program.

Note. SCCDP = Stepwise Clinical Competency Development Program; SBAR = Situation, Background, Assessment, Recommendation; CPR = Cardiopulmonary Resuscitation; ECG = Electrocardiogram.

**Fig 2. Overall structure and implementation of the SCCDP.** This figure depicts the overall structure and implementation of the SCCDP according to new nurses' tenure. The program consists of a basic stage and an advanced stage, each integrating lectures, hands-on practice, and simulation-based education. The basic stage focuses on foundational clinical skills and essential hospital competencies, while the advanced stage emphasizes system-based nursing care and emergency response through scenario-based simulations. Expert validation, structured feedback, and sequential delivery of educational sessions are incorporated throughout the program to support progressive competency development.

recruitment posters in each department. Participants were recruited through open enrollment. Before data collection, the purpose and procedures of the study were explained to the participants, and signed consent for voluntary participation was obtained. Data collection and program sessions were conducted in the hospital's designated training room and auditorium. A research assistant conducted a preliminary survey.

## Instruments

To assess changes over time in participants following their involvement in the SCCDP, their general characteristics, clinical knowledge (basic and advanced), clinical performance ability, and self-efficacy were measured. Each construct was assessed using validated instruments that were either newly developed or previously established in nursing education research. The clinical knowledge tool was developed by clinical nurse educators to align with the SCCDP content and validated through expert review, while the other instruments were well-established measures widely used in prior studies. All tools demonstrated acceptable levels of validity and reliability for use with new nurses. Detailed descriptions of each instrument are provided below.

## Clinical knowledge

The clinical knowledge assessment tool, developed by seven clinical nurse educators, was used to evaluate the knowledge of the main content covered in each session of the basic and advanced stages. This tool consisted of 30 questions at the basic level and 40 questions at the advanced level. The basic clinical knowledge assessment included seven sub-categories: intravenous injection (3 items), intradermal injection (3 items), Use of patient monitor and medical devices (5 items), Nursing care for maintaining and managing skin integrity (5 items), infection control and multidrug-resistant organisms (5 items), blood transfusion nursing care (5 items), and SBAR & handover (4 items). The advanced clinical knowledge assessment included four sub-categories: emergency system nursing (10 items), respiratory system nursing

(10 items), renal system nursing (10 items), and circulatory system nursing (10 items). Each item was scored 0 (incorrect) or 1 (correct), yielding total scores ranging from 0 to 30 (basic) and 0 to 40 (advanced), with higher scores indicating greater clinical knowledge. After development, the tool was validated by three nursing professors and three nursing managers with master's degrees or higher, resulting in a CVI of 1.0. The tool demonstrated a 95% confidence interval (CI) of 22.72–22.90 for the basic level and 25.79–27.35 for the advanced level. The reliability of the tool, expressed as a CI value, indicates the precision of the average estimate, with narrower intervals reflecting greater accuracy [25].

### Clinical performance ability

Clinical performance ability was assessed using the Korean Nursing Core Competency Measurement Tool (KNCCS) developed by Lee et al. [26]. This tool comprises 70 questions divided into subdomains: 12 questions on critical thinking, 21 questions on clinical performance, 13 questions on communication ability, 10 questions on understanding of humans, and 14 questions on professionalism and ethics. Responses are measured on a Likert scale ranging from 1 ("I cannot perform at all") to 5 ("I can perform with great confidence"), with higher scores indicating greater clinical performance ability. The tool's reliability at the time of development was Cronbach's $\alpha = .97$, and in this study, it was Cronbach's $\alpha = .98$.

### Self-efficacy

Self-efficacy was assessed using an instrument developed by Sherer et al. [27] and later modified and supplemented by Lee [28]. This tool comprises 17 items and is measured on a Likert scale ranging from 1 ("not at all") to 5 ("very much"). Higher scores indicate greater self-efficacy. The reliability of the tool was Cronbach's $\alpha = .94$ in Lee's study and Cronbach's $\alpha = .93$ in this study.

### Data analysis

The collected data were analyzed using SPSS for Windows (version 26.0; SPSS Inc., Chicago, IL, USA). Normality of the main outcome variables was assessed using the Kolmogorov–Smirnov test prior to inferential analysis. The general characteristics of the participants were analyzed using frequencies, percentages, means, and standard deviations. Variables that satisfied the normality assumption, including basic and advanced clinical knowledge and self-efficacy, were analyzed using repeated-measures analysis of variance (ANOVA). If the sphericity assumption was violated, the Greenhouse-Geisser correction was applied. Clinical performance ability variables did not meet the normality assumption and were therefore analyzed using Friedman's ANOVA. Cohen's effect sizes were calculated to determine the magnitude of the changes over time. Reliability of the measured variables was assessed using Cronbach's $\alpha$ coefficient and the 95% CI of the mean.

### Ethical considerations

This study was approved by the Chonnam National University Hospital Institutional Review Board (Approval No. CNUH-2023-256), and written informed consent was obtained from all participants. The voluntary nature, anonymity, and confidentiality of participation were explained to the participants, who were informed of their right to withdraw from the study at any time without penalty. Data confidentiality and anonymity were maintained throughout the study.

## Results

### General characteristics of the participants

The general characteristics of the participants are shown in Table 2. A total of 49 new nurses participated in this study, with an average age of 23.65 ± 1.95 years; 38 participants (77.6%) were under 24 years old, and 11 participants (22.4%) were between 24 and 30 years old. The majority of the participants were women (n = 37, 75.5%). Most participants were

**Table 2. The general characteristics of the participants (n = 49).**

| Characteristics | Categories | n (%) or M ± SD |
|---|---|---|
| Age (years) | ≤24 | 38 (77.6) |
| | >24~≤30 | 11 (22.4) |
| | Total | 23.65 ± 1.95 |
| Gender | Male | 12 (24.5) |
| | Female | 37 (75.5) |
| Marriage | Single | 48 (98.0) |
| | Married | 1 (2.0) |
| Department | Internal medicine ward | 17 (34.7) |
| | Surgical ward | 6 (12.2) |
| | Medical ICU | 3 (6.2) |
| | Surgical ICU | 5 (10.2) |
| | ER/OR | 18 (36.7) |

*Note.* M = Mean; SD = Standard Deviation; ICU = Intensive Care Unit; ER = Emergency Room; OR=Operating Room.

single (n = 48, 98.0%), with only one (2.0%) married. By department, the emergency and operating rooms had the highest representation with 18 participants (36.7%), followed by the internal medicine ward (n = 17, 34.7%), the surgical ward (n = 6, 12.2%), the surgical intensive care unit (n = 5, 10.2%), and the medical intensive care unit (n = 3, 6.2%).

### Effectiveness of the SCCDP for new nurses

The effectiveness of the SCCDP for new nurses is presented in Table 3. Basic clinical knowledge improved significantly immediately and three and six months after program participation, compared to pre-program levels ($F = 40.01$, $p < .001$). Among the subareas, significant improvements in medical device operation ($F = 72.63$, $p < .001$), infection control, multidrug-resistant organisms ($F = 26.28$, $p < .001$), and skin care ($F = 13.39$, $p = .004$) were observed. Notably, the area of medical device operation demonstrated the largest effect size (ES = 1.56–1.65). Conversely, no significant changes were observed in the SBAR and handover areas ($F = 3.80$, $p = .284$). Advanced clinical knowledge also showed significant improvement immediately and at three and six months after program participation, compared to pre-program levels ($F = 26.06$, $p < .001$). The greatest improvement was observed immediately after the SCCDP (ES = 1.84) and remained at a significantly high level three and six months after. Among the subdomains, emergency nursing ($F = 33.42$, $p < .001$), respiratory system nursing ($F = 29.61$, $p < .001$), renal system nursing ($F = 34.63$, $p < .001$), and circulatory system nursing ($F = 8.56$, $p < .001$), all showed significant improvements. The analysis for "Circulatory system nursing" includes 47 participants, excluding 2 who missed the lecture and related tasks. Emergency nursing particularly demonstrated the largest effect size immediately after the SCCDP (ES = 1.22). Clinical performance ability improved significantly three and six months following program participation compared to pre-program levels ($\chi^2 = 55.92$, $p < .001$). Significant improvements were observed across all subdomains, including interpersonal relationships and communication ($\chi^2 = 107.59$, $p < .001$), professional attitude ($\chi^2 = 77.57$, $p < .001$), critical thinking and evaluation ($\chi^2 = 43.67$, $p < .001$), general clinical performance ($\chi^2 = 35.51$, $p < .001$), and specialized clinical performance ($\chi^2 = 11.50$, $p < .001$). In particular, the interpersonal and communication demonstrated the largest effect size (ES = 2.05–2.09). Self-efficacy showed a slight increase at both three and six months following program participation compared to pre-program levels; however, the change was not statistically significant ($F = 2.80$, $p = .066$).

### Discussion

This study developed and implemented an SCCDP to improve the clinical performance of new nurses, support their effective adaptation to clinical settings, and increase their intention to remain in the workplace. To assess its effectiveness, the

**Table 3. Changes in the effectiveness of stepwise clinical competency development program for new nurses over time (n = 49).**

| Variables | Pre[a] Mean±SD | Post 1[b] Mean±SD | Post 2[c] Mean±SD | Post 3[d] Mean±SD | Source | F/χ² (p) | Effect size[a-b] (CI) | Effect size[a-c] (CI) | Effect size[a-d] (CI) |
|---|---|---|---|---|---|---|---|---|---|
| **Basic clinical knowledge** | 22.31±2.05 | 24.65±2.06 | 24.80±2.27 | 24.53±2.22 | Time | 40.01 (<.001)§ Post hoc: a<b,c,d | 1.14 (0.71-1.58) | 1.15 (0.72-1.59) | 1.04 (0.61-1.47) |
| Intravenous injection | 2.33±0.63 | 2.27±0.73 | 2.53±0.62 | 2.51±0.58 | Time | 9.69 (.021)§ | −0.01 (−0.49-0.32) | 0.32 (−0.09-0.73) | 0.30 (−0.10-0.70) |
| Intradermal injection | 2.57±0.54 | 2.37±0.60 | 2.71±0.46 | 2.69±0.47 | Time | 14.91 (.002)§ Post hoc: b<d | −0.34 (−0.75-0.07) | 0.27 (−0.14-0.68) | 0.23 (−0.18-0.64) |
| Use of patient monitor and medical devices | 3.02±0.66 | 4.12±0.75 | 4.23±0.80 | 4.06±0.78 | Time | 72.63 (<.001)§ Post hoc: a<b,c,d | 1.56 (1.10-2.02) | 1.65 (1.18-2.12) | 1.44 (0.99-1.89) |
| Nursing care for maintaining and managing skin integrity | 3.65±0.97 | 4.23±0.84 | 3.74±1.10 | 3.71±1.04 | Time | 13.39 (.004)§ Post hoc: a<b, b>d | 0.64 (0.23-1.05) | 0.09 (−0.32-0.49) | 0.06 (−0.35-0.46) |
| Infection control & multidrug-resistant organisms | 2.98±0.99 | 3.61±0.81 | 3.88±0.83 | 3.78±0.92 | Time | 26.28 (<.001)§ Post hoc: a<b,c,d | 0.70 (0.28-1.11) | 0.99 (0.56-1.41) | 0.84 (0.42-1.26) |
| Blood transfusion nursing care | 4.63±0.53 | 4.76±0.43 | 4.43±0.71 | 4.53±0.54 | Time | 11.59 (.009)§ Post hoc: b>c | 0.27 (−0.14-0.68) | −0.32 (−0.73-0.09) | −0.19 (−0.59-0.22) |
| SBAR & handover | 3.12±0.53 | 3.27±0.45 | 3.29±0.50 | 3.25±0.52 | Time | 3.80 (.284)§ | 0.31 (−0.10-0.71) | 0.33 (−0.08-0.74) | 0.25 (−0.16-0.65) |
| **Advanced clinical knowledge** | 26.57±2.72 | 31.86±3.01 | 29.12±3.78 | 28.53±2.84 | Time | 26.06 (<.001) Post hoc: a<b,c,d, b>c,d | 1.84 (1.36-2.33) | 0.77 (0.36-1.19) | 0.71 (0.29-1.12) |
| Emergency system nursing | 6.67±1.31 | 8.18±1.17 | 7.00±1.31 | 6.71±1.41 | Time | 33.42 (<.001)§ Post hoc: a<b, b>c,d | 1.22 (0.78-1.66) | 0.25 (−0.15-0.66) | 0.03 (−0.38-0.43) |
| Respiratory system nursing | 6.29±1.44 | 7.74±1.10 | 7.08±1.24 | 7.00±1.10 | Time | 29.61 (<.001)§ Post hoc: a<b, b>c,d | 1.13 (0.70-1.57) | 0.59 (0.18-1.00) | 0.55 (0.14-0.97) |
| Renal system nursing | 6.65±1.11 | 8.14±1.06 | 7.31±1.50 | 6.92±1.30 | Time | 34.63 (<.001)§ Post hoc: a<b, b>c,d | 1.37 (0.92-1.82) | 0.50 (0.09-0.91) | 0.22 (−0.18-0.63) |
| Circulatory system nursing* | 6.94±1.74 | 8.13±1.39 | 7.68±1.42 | 7.85±1.18 | Time | 8.56 (<.001)£ Post hoc: a<b,d | 0.76 (0.34-1.17) | 0.47 (0.06-0.88) | 0.61 (0.20-1.03) |
| **Clinical performance ability** | 2.94±0.57 | – | 3.82±0.46 | 3.82±0.48 | Time | 55.92 (<.001)§ Post hoc: a<c,d | – | 1.70 (1.23-2.17) | 1.67 (1.20-2.14) |
| Interpersonal relationships and communication | 2.67±0.63 | – | 3.81±0.47 | 3.84±0.48 | Time | 107.59 (<.001)£ Post hoc: a<c,d | – | 2.05 (1.55-2.55) | 2.09 (1.59-2.59) |
| Professional attitude | 2.57±0.66 | – | 3.63±0.58 | 3.69±0.53 | Time | 77.57 (<.001)£ Post hoc: a<c,d | | 1.71 (1.23-2.18) | 1.87 (1.39-2.36) |
| Critical thinking and evaluation | 3.22±0.65 | – | 3.93±0.46 | 3.88±0.50 | Time | 43.67 (<.001)§ Post hoc: a<c,d | | 1.26 (0.82-1.70) | 1.34 (0.70-1.57) |
| General clinical performance | 3.17±0.69 | – | 3.90±0.50 | 3.87±0.53 | Time | 35.51 (<.001)§ Post hoc: a<c,d | | 1.21 (0.77-1.65) | 1.14 (0.70-1.57) |
| Specialized clinical performance | 3.34±0.68 | – | 3.81±0.61 | 3.82±0.66 | Time | 11.50 (<.001)£ Post hoc: a<c,d | | 0.73 (0.31-1.15) | 0.72 (0.30-1.13) |
| **Self-efficacy** | 65.47±8.97 | – | 68.57±9.94 | 68.04±10.40 | Time | 2.80 (.066) | | 0.33 (−0.08-0.73) | 0.27 (−0.14-0.67) |

*Note.* Pre = Pre-program; Post 1 = Immediately after program; Post 2 = 3 months after program; Post 3 = 6 months after program; SD = Standard deviation; SBAR = Situation, Background, Assessment, Recommendation; §Friedman's ANOVA; £Greenhouse-Geisser; *Although 49 participants were registered in the study, only 47 were included in the analysis for "Circulatory system nursing" because 2 participants did not attend the circulatory system lecture due to scheduling conflicts.

SCCDP was structured into basic and advanced stages and applied over time. Clinical knowledge, clinical performance ability, and self-efficacy were evaluated to confirm the SCCDP's effectiveness. The findings demonstrated significant improvements in both basic and advanced clinical knowledge and clinical performance ability following program participation. Importantly, these effects were sustained immediately after the SCCDP and at three- and six-month follow-up assessments. These findings suggest that structured, longitudinal competency-based education can effectively bridge the gap between initial orientation and ongoing professional development for new nurses.

The SCCDP provided an integrated framework that combined theoretical lectures, practical training, and simulations, effectively linking classroom knowledge to real-world performance. Consistent with previous reviews showing that structured residency programs enhance confidence and retention among new nurses [5,7,8], this study extends prior evidence by demonstrating that a staged, competency-based approach supports both the technical and interpersonal dimensions of nursing competence over time. Such findings emphasize that structured, repetitive learning with feedback allows nurses to consolidate and retain clinical competencies that are directly applicable to complex practice settings.

Education led by clinical nurse educators has been positively evaluated before, as it enables systematic and level-specific instructions, which has been proven to reduce anxiety among new nurses [11,29]. New nurses often experience psychological anxiety due to limited practical skills, gaps between theoretical knowledge and clinical practice, and pressure to perform competently [30]. Furthermore, improper education that does not teach the appropriate skill levels may widen this gap. To address this, retention rates may be improved by providing training that combines practice and simulations tailored to the developmental and adaptation stages of new nurses [16]. In addition, the gap between the theoretical knowledge required at each stage and practical applications in clinical settings can be minimized, and simulation practice based on real cases can enhance learning confidence while reinforcing critical thinking skills [31]. This aligns with meta-analytic evidence indicating that simulation-based education is particularly effective when aligned with learners' developmental needs [32]. Our results further support the need for stepwise education for new nurses engaged in intensive work, as suggested by prior Korean and international studies [9,29,33]. In particular, the sustained improvement in medical device operation and emergency nursing competencies emphasizes that structured repetition and simulation contribute to safer clinical practice. To achieve broader impact, future applications of the SCCDP should be tested in varied hospital settings and compared with existing residency programs to evaluate scalability and standardization.

The results of this study revealed that the basic clinical knowledge of new nurses significantly improved immediately and three and six months after the SCCDP. Their ability to operate medical devices improved the most. These suggest that the new nurses learned the correct usage of various medical devices they handle in the field, enhancing their ability to operate them safely. Of the participants, 53.0% worked primarily in emergency rooms, operating rooms, and intensive care units, where they frequently used various high-risk medical devices. Through repeated use, they are likely to become more proficient in operating these devices over time, thereby contributing to improved clinical knowledge. New nurses with one to three months of experience were found to frequently make mistakes and face difficulties when using unfamiliar medical devices [9], with a high incidence of safety accidents related to these devices [34]. Since the use of inappropriate equipment can result in patient safety incidents, it is essential to provide practical training on equipment use at the start of employment. As this is a critical competency for enhancing patient safety and improving nurses' work efficiency, it should become an integral component of the initial clinical adaptation process for new nurses. It is also crucial to develop a simulation program based on error events that occur in real clinical settings. These findings suggest that repeated exposure to simulation-based learning and feedback not only enhances psychomotor proficiency but also strengthens nurses' confidence in managing high-risk situations, thereby contributing to safer patient care. The sustained improvement observed over time further indicates that the SCCDP's progressive structure and ongoing evaluation promoted long-term learning retention—an element often missing in short-term orientation programs.

Advanced clinical knowledge was also sustained over time, with the greatest impact in emergency nursing. This indicates that the new nurses effectively acquired the specialized skills and knowledge required for emergency situations,

significantly enhancing their ability to respond to emergencies in real clinical settings. Since nurses serve as the first responders in emergency situations, they require complex competencies, including specialized knowledge, quick judgment, and strong decision-making skills, to respond effectively [33,35]. Emergency nursing is one of the most challenging areas frequently encountered by new nurses. A lack of experience in handling emergency situations can reduce confidence in patient care. Therefore, it is crucial to provide education that incorporates simulation-based training [35]. Experience in responding to emergency situations provides a sense of security and confidence as it helps individuals understand the complexities of treatments and procedures in emergency contexts [36]. The findings reveal that the educational program enhanced new nurses' understanding of emergency nursing, developed their ability to prevent the worsening of patients' conditions, and delivered effective emergency care through simulation exercises.

The results of this study demonstrate that the greatest effect among clinical performance abilities was observed in interpersonal relationships and communication. This indicates that the SCCDP enhanced nurses' ability to collaborate effectively within the team through clear communication with patients, families, and other medical staff. Nurses spend the most time with patients, making interpersonal and communication skills essential competencies. Therefore, education aimed at improving these skills is necessary [37,38]. Communication competency is considered an essential skill for nurses to provide safe patient-centered care in clinical settings. However, no significant changes were observed in the SBAR and handover capabilities. Handover is a critical process for exchanging information between nurses and an important factor that can impact patient care. SBAR-based handovers were found to enhance empathy and trust between nurses and patients and improve responsiveness to questions and requests from patients or their families, thereby strengthening communication among medical staff as well [39]. However, this difference in findings may be because the time allocated to intensive education in this area was relatively insufficient. New nurses find handovers challenging and often feel because of their limited ability to conduct them efficiently [40,41]. To maximize knowledge and skills and enhance competency and confidence, repeated education, simulation training, and a continuous feedback system should be provided. The improvement observed in communication and interpersonal relationships underscores the SCCDP's effectiveness in cultivating collaboration and teamwork—competencies that directly influence patient safety and care quality. The integration of SBAR and team-based simulation provided opportunities for nurses to rehearse communication in realistic contexts, reinforcing both relational and clinical proficiency.

Self-efficacy showed a slight increase following the SCCDP; however, this change was not statistically significant. This finding contrasts with prior reports where structured mentorship programs significantly enhanced self-efficacy [42]. It is possible that the SCCDP, while effective for technical and performance outcomes, requires supplementation with longitudinal mentoring and psychosocial support to influence psychological variables. Additionally, self-efficacy is associated with an individual's positive response to challenges [43], which may explain why the increasing workload and high role expectations experienced by new nurses contribute to an increased sense of burden over time [44]. For new nurses, the transition period from student to nurse is critical for professional growth and development. To enhance self-efficacy through positive experiences and appropriate support, continuous mentoring, accumulation of successful experiences, and psychological support are needed alongside educational programs [9,42]. This suggests that future adaptation programs should integrate structured mentorship and reflective coaching components to enhance psychological resilience and confidence in addition to technical competence.

In summary, this study contributes evidence that a structured, competency-based, and stepwise educational program improves new nurses' knowledge and clinical performance, aligning with international calls for standardized adaptation programs. However, it has several limitations. First, additional educational interventions are required to further develop SBAR and handover skills. Second, educational strategies that incorporate long-term mentoring and continuous feedback are necessary to enhance psychological competencies such as self-efficacy. Third, since our study used a convenient sample from a single medical institution, caution must be exercised when generalizing the findings. Moreover, simultaneous administration of lengthy instruments may have induced survey fatigue, potentially biasing responses.

Future research should pursue three main directions to strengthen the evidence base for the SCCDP and expand its applicability. First, controlled or randomized trial designs are needed to rigorously evaluate causal effects of SCCDP participation. Second, multicenter and cross-national studies should be conducted to evaluate its generalizability and scalability across diverse institutional settings. Third, mixed-methods or qualitative follow-up studies may provide nuanced insights into the experiential and organizational factors influencing adaptation, self-efficacy, and retention. Such evidence will help refine competency-based programs and establish them as a standardized strategy for supporting new nurses' transition to practice.

## Conclusion

The SCCDP developed in this study combines theoretical content, practical training, and simulations. This effectively improved the basic and advanced clinical knowledge and clinical performance abilities of new nurses, and these improvements were sustained over time. This program provides an opportunity to bridge the gap between theoretical knowledge and practical applications, enabling nurses to effectively apply the learned concepts in real clinical settings. In the long term, the SCCDP helped new nurses to better adapt to the clinical environment. Regarding sustainability, the SCCDP was systematically integrated into the existing hospital education structure, utilizing internal resources such as clinical nurse educators and standardized training protocols, making the SCCDP feasible and sustainable for long-term implementation. Additionally, this study highlights the necessity of structured support programs that help new nurses effectively integrate and apply their theoretical knowledge within complex clinical settings, thereby facilitating a smoother transition from academic learning to professional practice. The SCCDP developed in this study can be utilized to enhance the clinical adaptation and competency of new nurses, potentially contributing to reducing nurse turnover and providing safer and more effective nursing care in clinical settings.

## Acknowledgments

The authors express their heartfelt appreciation to the participants and thank them for their valuable time and support.

## Author contributions

**Conceptualization:** Hye Won Jeong.

**Data curation:** Shinhye Ahn.

**Formal analysis:** Hye Won Jeong.

**Funding acquisition:** Hye Won Jeong.

**Investigation:** Shinhye Ahn.

**Methodology:** Hye Won Jeong.

**Project administration:** Hye Won Jeong.

**Resources:** Shinhye Ahn.

**Software:** Hye Won Jeong.

**Supervision:** Hye Won Jeong.

**Validation:** Hye Won Jeong.

**Visualization:** Shinhye Ahn.

**Writing – original draft:** Shinhye Ahn, Hye Won Jeong.

**Writing – review & editing:** Shinhye Ahn, Hye Won Jeong.

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
