## [Decision Letter · Decision Letter 0]

2 Apr 2025

PONE-D-25-10399Development and evaluation of a stepwise clinical competency development program for new nurses: A single-group repeated-measures quasi-experimental studyPLOS ONE

Dear Dr. Jeong,

Thank you for submitting your manuscript to PLOS ONE. After careful consideration, we feel that it has merit but does not fully meet PLOS ONE’s publication criteria as it currently stands. Therefore, we invite you to submit a revised version of the manuscript that addresses the points raised during the review process.

We look forward to receiving your revised manuscript.

Kind regards,

Joyce Jebet Cheptum

Academic Editor

PLOS ONE

“This study was funded by the National Research Foundation of Korea (NRF-2022R1F1A1067574).”

Reviewers' comments:

Reviewer's Responses to Questions

**Comments to the Author**

1. Is the manuscript technically sound, and do the data support the conclusions?

Reviewer #1: Yes

Reviewer #2: Yes

2. Has the statistical analysis been performed appropriately and rigorously? 

Reviewer #1: Yes

Reviewer #2: Yes

3. Have the authors made all data underlying the findings in their manuscript fully available?

Reviewer #1: Yes

Reviewer #2: Yes

4. Is the manuscript presented in an intelligible fashion and written in standard English?

Reviewer #1: Yes

Reviewer #2: Yes

5. Review Comments to the Author

Reviewer #1: Please provide the following

1. operational definition of new nurses

2. citation for padlet platform, a short insight into it

3. role of investigator is missing in study as it is not specified who provided the lectures/practice/simulations and who took feedbacks. how many people were involved? whether there is slight chance of error

4. is endotracheal intubation a nurse led procedure or nurses' assist in that?

Reviewer #2: Overall, a well developed detailed manuscripts. The theme of teh research shows relevancy to the readers and need of the time. I myself being a nurse educator totally agree to implementation of bootcamps, trainings and orientations for novice nurses and allied health. However, there is a clarity required as to 3 questionnaires of clinical knowledge, performance and self efficacy were used all together before, immediate and after training. The training modules doesnot have ECG machine, Cardiac monitor, Pharmacology, leadership, critical thinking, basic nursing skills such as Nasogastric insertion, feeding, catheterization topics in it. Training modules need sequence may b elater on, It will be effective to use controlled and uncontrolled group methodology to see effectiveness of the training perhaps later on stages it can be developed. Does participants got exhausted with lengthy questionnaires, it could be a limitation.

P.S; Also would like to work with the authors and team on the same projects

6. PLOS authors have the option to publish the peer review history of their article (what does this mean? ). If published, this will include your full peer review and any attached files.

**Do you want your identity to be public for this peer review?** For information about this choice, including consent withdrawal, please see our Privacy Policy .

Reviewer #1: **Yes:** GEETANJALI LOOMBA

Reviewer #2: **Yes:** Zaibunissa

---

## [Author Response · Author response to Decision Letter 1]

10 Apr 2025

We sincerely thank the reviewers for their insightful and constructive feedback on our manuscript. We have carefully addressed each comment raised by the reviewers and have revised the manuscript accordingly.

Our detailed responses to each comment, including specific revisions made, have been provided in the uploaded response documents titled "Response to Reviewer #1" and "Response to Reviewer #2." We kindly ask you to refer to these documents for comprehensive details regarding the modifications made in response to the reviewers' valuable suggestions.

We appreciate the opportunity to further enhance the quality of our manuscript and hope that the revisions meet your expectations.

Thank you again for your consideration.

Sincerely,

---

## [Decision Letter · Decision Letter 1]

8 Jul 2025

PONE-D-25-10399R1Development and evaluation of a stepwise clinical competency development program for new nurses: A single-group repeated-measures quasi-experimental studyPLOS ONE

Dear Dr. Jeong,

Thank you for submitting your manuscript to PLOS ONE. After careful consideration, we feel that it has merit but does not fully meet PLOS ONE’s publication criteria as it currently stands. Therefore, we invite you to submit a revised version of the manuscript that addresses the points raised during the review process.

Dear Author,

Your research is valuable, and the following comments may help enhance your work:

1. The necessity of conducting this research is not clearly defined. Please provide a clearer rationale for your study.

2. In the methods section, clarify why the study was conducted as a quasi-experimental study.

3. At the end of the discussion, please include suggestions for future research.

4. After the conclusion, declare any conflicts of interest related to the study.

We look forward to receiving your revised manuscript.

Kind regards,

Joyce Jebet Cheptum

Academic Editor

PLOS ONE

Journal Requirements:

Additional Editor Comments:

Dear Author,

Please work on the comments in totality.

Reviewers' comments:

Reviewer's Responses to Questions

**Comments to the Author**

1. If the authors have adequately addressed your comments raised in a previous round of review and you feel that this manuscript is now acceptable for publication, you may indicate that here to bypass the “Comments to the Author” section, enter your conflict of interest statement in the “Confidential to Editor” section, and submit your "Accept" recommendation.

Reviewer #3: (No Response)

Reviewer #4: (No Response)

2. Is the manuscript technically sound, and do the data support the conclusions?

Reviewer #3: (No Response)

Reviewer #4: Yes

3. Has the statistical analysis been performed appropriately and rigorously? 

Reviewer #3: (No Response)

Reviewer #4: Yes

4. Have the authors made all data underlying the findings in their manuscript fully available?

Reviewer #3: (No Response)

Reviewer #4: No

5. Is the manuscript presented in an intelligible fashion and written in standard English?

Reviewer #3: (No Response)

Reviewer #4: Yes

6. Review Comments to the Author

Reviewer #4: The manuscript depicts a significance area in nursing practice. It is well written in standard english.

The areas of clarification include

1.iIs it that the nurses arent competent at employment? is the problem in the curriculum and how they are trained? 2. In the background- What is the implication of these turnover and lack of competences to patient outcomes? Any evidence?

3. The training programs development- were they validated? by who? what approach was used to validate them? What informed the content?

4. Were the nurses doing this as they continued working? if so is the effect due to the training or the hands on experience they gained as they worked?

5. Those nurses who were less than a year at the hospital but had prior experience were they included?

The content in table 1 - the areas of training is similar to what is taught in school to nurses- Is this to say they dont have these skills at employment?

6. The period from graduation to employment and prior experience may have varied, and this has potential to vary their competencies Was this considered in the inclusion into the programme?

7. The results are highly summarised loosing the interpretations from the various tools with various items . The key themes of training with their specific tools could be presented as independ tables

6. Is the programe sustainable? scalable? Whats the generazability of the study findings?

7. Any study recommendations?

7. PLOS authors have the option to publish the peer review history of their article (what does this mean? ). If published, this will include your full peer review and any attached files.

**Do you want your identity to be public for this peer review?** For information about this choice, including consent withdrawal, please see our Privacy Policy .

Reviewer #3: No

Reviewer #4: **Yes:** Rosemary Kawira Kithuci

---

## [Author Response · Author response to Decision Letter 2]

19 Jul 2025

Thank you very much for providing the opportunity to revise our manuscript.

We have carefully considered and addressed all comments raised by the reviewers.

Please find attached our detailed responses to the reviewers' comments.

Thank you once again for your valuable feedback and consideration.

---

## [Decision Letter · Decision Letter 2]

15 Aug 2025

PONE-D-25-10399R2Development and evaluation of a stepwise clinical competency development program for new nurses: A single-group repeated-measures quasi-experimental studyPLOS ONE

Dear Dr. Jeong,

Thank you for submitting your manuscript to PLOS ONE. After careful consideration, we feel that it has merit but does not fully meet PLOS ONE’s publication criteria as it currently stands. Therefore, we invite you to submit a revised version of the manuscript that addresses the points raised during the review process.

We look forward to receiving your revised manuscript.

Kind regards,

Joyce Jebet Cheptum

Academic Editor

PLOS ONE

Journal Requirements:

Reviewers' comments:

Reviewer's Responses to Questions

**Comments to the Author**

1. If the authors have adequately addressed your comments raised in a previous round of review and you feel that this manuscript is now acceptable for publication, you may indicate that here to bypass the “Comments to the Author” section, enter your conflict of interest statement in the “Confidential to Editor” section, and submit your "Accept" recommendation.

Reviewer #3: (No Response)

Reviewer #4: All comments have been addressed

2. Is the manuscript technically sound, and do the data support the conclusions?

Reviewer #3: (No Response)

Reviewer #4: Yes

3. Has the statistical analysis been performed appropriately and rigorously? 

Reviewer #3: (No Response)

Reviewer #4: Yes

4. Have the authors made all data underlying the findings in their manuscript fully available?

Reviewer #3: (No Response)

Reviewer #4: Yes

5. Is the manuscript presented in an intelligible fashion and written in standard English?

Reviewer #3: (No Response)

Reviewer #4: Yes

6. Review Comments to the Author

Reviewer #3: Dear Author,

Thank you for your valuable research. However, I have a few comments that could help enhance your work:

1. In the abstract, the necessity of the study is not clearly articulated. Please make this more explicit.

2. The last paragraph of the introduction should address both the necessity of the research and the innovation it brings.

3. In the methods section, please provide a comprehensive explanation of the tools used for data collection, including a discussion of their reliability.

4. The inclusion and exclusion criteria for the study are incomplete. Ensure that these criteria are fully detailed.

5. Please strengthen the discussion section. Provide an analysis of the studies, and at the end, offer suggestions for future research.

6. Was there a conflict of interest in this study? If so, please disclose it.

7. All abbreviations should be defined and listed at the end of the study.

I hope these suggestions are helpful in refining your manuscript.

Reviewer #4: The author has addressed the comments satisfactorily especially the key one on the intervention development and validation process.

Consider a remaining the conceptual framework a theoretical framework - you are emebding the study on CBE model

Does the study have any recommendation to training institutitions? or clinical learning among students within your hospital? - considering you had to train for soo many hours after graduating from college- Are there deficits from Curricula? Is clinical teaching and learning for nursing students deficient?

Separate the sustainability and generazability of the findings from conclusion- Have it as a subheading below it Limitations of the study can be included and allude to the generazability and scalability of the findings

7. PLOS authors have the option to publish the peer review history of their article (what does this mean? ). If published, this will include your full peer review and any attached files.

**Do you want your identity to be public for this peer review?** For information about this choice, including consent withdrawal, please see our Privacy Policy .

Reviewer #3: No

Reviewer #4: **Yes:** Rosemary Kawira Kithuci

---

## [Author Response · Author response to Decision Letter 3]

16 Aug 2025

We have attached files containing our detailed responses to all reviewer comments.

---

## [Decision Letter · Decision Letter 3]

29 Sep 2025

PONE-D-25-10399R3Development and evaluation of a stepwise clinical competency development program for new nurses: A single-group repeated-measures quasi-experimental studyPLOS ONE

Dear Dr. Jeong,

Thank you for submitting your manuscript to PLOS ONE. After careful consideration, we feel that it has merit but does not fully meet PLOS ONE’s publication criteria as it currently stands. Therefore, we invite you to submit a revised version of the manuscript that addresses the points raised during the review process.

We look forward to receiving your revised manuscript.

Kind regards,

Joyce Jebet Cheptum

Academic Editor

PLOS ONE

Journal Requirements:

Additional Editor Comments:

Enhance the methodology section on the abstract

Reviewers' comments:

Reviewer's Responses to Questions

**Comments to the Author**

1. If the authors have adequately addressed your comments raised in a previous round of review and you feel that this manuscript is now acceptable for publication, you may indicate that here to bypass the “Comments to the Author” section, enter your conflict of interest statement in the “Confidential to Editor” section, and submit your "Accept" recommendation.

Reviewer #1: All comments have been addressed

Reviewer #3: (No Response)

2. Is the manuscript technically sound, and do the data support the conclusions?

Reviewer #1: Yes

Reviewer #3: (No Response)

3. Has the statistical analysis been performed appropriately and rigorously? 

Reviewer #1: Yes

Reviewer #3: (No Response)

4. Have the authors made all data underlying the findings in their manuscript fully available?

Reviewer #1: Yes

Reviewer #3: (No Response)

5. Is the manuscript presented in an intelligible fashion and written in standard English?

Reviewer #1: Yes

Reviewer #3: (No Response)

6. Review Comments to the Author

Reviewer #1: (No Response)

Reviewer #3: Dear Author,

Your research is valuable; however, the following comments may help enhance it:

1. The necessity of the research is not clearly articulated in the abstract.

2. In the introduction, please highlight the innovation of your research in the last paragraph.

3. The methodology outlined in the abstract is incomplete.

4. The methodology section should provide a comprehensive explanation of the data collection tool used.

5. Consider strengthening the discussion section for greater impact.

Best regards.

7. PLOS authors have the option to publish the peer review history of their article (what does this mean? ). If published, this will include your full peer review and any attached files.

**Do you want your identity to be public for this peer review?** For information about this choice, including consent withdrawal, please see our Privacy Policy .

Reviewer #1: No

Reviewer #3: No

---

## [Author Response · Author response to Decision Letter 4]

10 Nov 2025

Dear reviewer,

We have carefully revised the manuscript according to all reviewer comments.

All issues raised by the reviewers have been fully addressed, and the revised files (clean version and tracked-changes version) are attached for your consideration.

Thank you for your time and kind attention.

Sincerely,

Hye-Won Jeong

---

## [Decision Letter · Decision Letter 4]

21 Dec 2025

PONE-D-25-10399R4Development and evaluation of a stepwise clinical competency development program for new nurses: A single-group repeated-measures quasi-experimental studyPLOS One

Dear Dr. Jeong,

Thank you for submitting your manuscript to PLOS ONE. After careful consideration, we feel that it has merit but does not fully meet PLOS ONE’s publication criteria as it currently stands. Therefore, we invite you to submit a revised version of the manuscript that addresses the points raised during the review process.

We look forward to receiving your revised manuscript.

Kind regards,

Joyce Jebet Cheptum

Academic Editor

PLOS One

Journal Requirements:

Additional Editor Comments:

There are a few more comments to address

Reviewers' comments:

Reviewer's Responses to Questions

**Comments to the Author**

1. If the authors have adequately addressed your comments raised in a previous round of review and you feel that this manuscript is now acceptable for publication, you may indicate that here to bypass the “Comments to the Author” section, enter your conflict of interest statement in the “Confidential to Editor” section, and submit your "Accept" recommendation.

Reviewer #2: All comments have been addressed

Reviewer #5: All comments have been addressed

2. Is the manuscript technically sound, and do the data support the conclusions?

Reviewer #2: Yes

Reviewer #5: Yes

3. Has the statistical analysis been performed appropriately and rigorously? 

Reviewer #2: Yes

Reviewer #5: Yes

4. Have the authors made all data underlying the findings in their manuscript fully available?

Reviewer #2: Yes

Reviewer #5: Yes

5. Is the manuscript presented in an intelligible fashion and written in standard English?

Reviewer #2: (No Response)

Reviewer #5: Yes

6. Review Comments to the Author

Reviewer #2: Very detailed and easy to understand the research and implement, looking forward to more studies in this area and further collaboration with researchers

Reviewer #5: Reviewer Report

The manuscript titled “Development and evaluation of a stepwise clinical competency development program for new nurses: A single-group repeated-measures quasi-experimental study” addresses an important topic of supporting new nurses’ transition to practice using a structured, competency-based educational program. The longitudinal evaluation and integration of competency-based education principles are commendable. However, few issues require attention:

1. Statistical Analysis

Statistical analysis has been partially met as it is generally appropriate for the study design and research objectives. The use of a single-group repeated-measures quasi-experimental design is clearly stated and justified given ethical and institutional constraints. The authors correctly employed repeated measures ANOVA for normally distributed data and Friedman’s ANOVA for non-parametric data with effect sizes reported to support interpretation of magnitude. However, some aspects of rigor and transparency remain insufficiently addressed for instance;

• Although the manuscript states that normality was tested using the Kolmogorov–Smirnov test, the results of these tests are not reported nor is it specified which variables violated normality assumptions and therefore required non-parametric analysis.

• The rationale for switching between parametric and non-parametric tests would be strengthened by clearly linking normality test outcomes to the selected analyses.

• Reporting exact p-values for assumption testing or summarizing them in a supplementary table would improve methodological transparency.

• The use of Greenhouse–Geisser correction is appropriate, but the manuscript would benefit from explicitly stating which outcomes required this correction

2. Presentation

The manuscript is generally well organized, clearly structured and written in understandable standard academic English. The background, methods, results, and discussion sections are logically sequenced and easy to follow. Technical terminology is used appropriately and consistently.

However, minor language issues persist, including:

• Occasional redundancy in the Discussion section, particularly where key findings are restated multiple times.

• Some lengthy sentences that could be simplified to improve readability.

• Minor grammatical and stylistic inconsistencies (e.g., tense use and article usage).

These issues do not impede comprehension but would benefit from final professional language polishing.

3. Methodology

The manuscript is technically sound. The study design, instruments, intervention structure, and outcome measures are clearly described and appropriate for the research aims. Validated tools with strong reliability coefficients were used, and the intervention was systematically implemented over an adequate follow-up period.

The data presented support the authors’ main conclusions that the Stepwise Clinical Competency Development Program (SCCDP) improved clinical knowledge and clinical performance ability among new nurses. The lack of significant change in self-efficacy is appropriately acknowledged and discussed, demonstrating balanced interpretation.

Limitations including the absence of a control group and single-center design are transparently discussed and do not undermine the internal validity of the findings. The conclusions are cautious, proportionate, and aligned with the data presented.

4. Final Comment

This manuscript represents a well-developed and methodologically sound evaluation of a structured competency-based training program for new nurses. The longitudinal follow-up and use of validated instruments are particular strengths, and the topic is relevant to healthcare workforce development.

To further strengthen the manuscript, I recommend:

1. Enhancing the Data Analysis section by explicitly reporting normality test outcomes and clearly stating which variables required non-parametric analysis.

2. Reduce minor redundancy in the Discussion and perform a final language polish.

3. Ensure consistency in terminology and formatting throughout the manuscript.

I have no concerns regarding research ethics, data integrity, or dual publication. Ethical approval and informed consent are clearly documented, and the study adheres to accepted publication standards.

7. PLOS authors have the option to publish the peer review history of their article (what does this mean? ). If published, this will include your full peer review and any attached files.

**Do you want your identity to be public for this peer review?** For information about this choice, including consent withdrawal, please see our Privacy Policy .

Reviewer #2: **Yes:** Zaibunissa

Reviewer #5: **Yes:** DR. RACHEAL MUKOYA MASIBO

---

## [Author Response · Author response to Decision Letter 5]

2 Jan 2026

Dear Reviewer,

Thank you for your careful review and constructive comments on our manuscript.

All reviewer comments have been addressed, and the manuscript has been revised accordingly.

Specifically, the Data Analysis section was clarified, minor redundancy in the Discussion was reduced,

and terminology and formatting were standardized throughout the manuscript.

Detailed, point-by-point responses are provided in the attached response document.

We appreciate your valuable feedback and consideration.

Sincerely,

The Authors

---

## [Decision Letter · Decision Letter 5]

26 Jan 2026

Development and evaluation of a stepwise clinical competency development program for new nurses: A single-group repeated-measures quasi-experimental study

PONE-D-25-10399R5

Dear Dr. Jeong,

We’re pleased to inform you that your manuscript has been judged scientifically suitable for publication and will be formally accepted for publication once it meets all outstanding technical requirements.

Kind regards,

Joyce Jebet Cheptum

Academic Editor

PLOS One

Additional Editor Comments (optional):

Reviewers' comments:

Reviewer's Responses to Questions

**Comments to the Author**

1. If the authors have adequately addressed your comments raised in a previous round of review and you feel that this manuscript is now acceptable for publication, you may indicate that here to bypass the “Comments to the Author” section, enter your conflict of interest statement in the “Confidential to Editor” section, and submit your "Accept" recommendation.

Reviewer #5: All comments have been addressed

2. Is the manuscript technically sound, and do the data support the conclusions?

Reviewer #5: Yes

3. Has the statistical analysis been performed appropriately and rigorously? 

Reviewer #5: Yes

4. Have the authors made all data underlying the findings in their manuscript fully available?

Reviewer #5: Yes

5. Is the manuscript presented in an intelligible fashion and written in standard English?

Reviewer #5: Yes

6. Review Comments to the Author

Reviewer #5: The manuscript has been substantially improved and is methodologically sound, well written, and suitable for publication ]The authors clearly attempted to address the reviewer’s concerns, particularly regarding clarification of data analysis procedures, discussion refinement, and terminology consistency.

As a minor optional refinement, the manuscript would benefit from a final professional edit to address occasional issues such as consistent and context-appropriate use of “the” and “a” and to correct minor tense shifts in the Discussion Section; these are purely editorial adjustments and do not affect the study’s methodology, analysis, or overall scientific validity.

7. PLOS authors have the option to publish the peer review history of their article (what does this mean? ). If published, this will include your full peer review and any attached files.

**Do you want your identity to be public for this peer review?** For information about this choice, including consent withdrawal, please see our Privacy Policy .

Reviewer #5: **Yes:** Dr Racheal Masibo

---

## [Editor Report · Acceptance letter]

2 April 2025

PONE-D-25-10399R5

PLOS One

Dear Dr. Jeong,

I'm pleased to inform you that your manuscript has been deemed suitable for publication in PLOS One. Congratulations! Your manuscript is now being handed over to our production team.

Kind regards,

on behalf of

Dr. Joyce Jebet Cheptum

Academic Editor

PLOS One